

# Pore morphology of polar firn around closure revealed by X-ray tomography

Alexis Burr[1,2], Clément Ballot[1,2], Pierre Lhuissier[2], Patricia Martinerie[1], Christophe L. Martin[2], and Armelle Philip[1]

[1]Univ. Grenoble Alpes, Grenoble INP*, CNRS, IRD, IGE, 38000 Grenoble, France
[2]Univ. Grenoble Alpes, CNRS, Grenoble INP*, SIMaP, 38000 Grenoble, France

*Correspondence to:* Armelle.Philip@univ-grenoble-alpes.fr

**Abstract.** Understanding the slow densification process of polar firn into ice is essential in order to constrain the age difference between the ice matrix and entrapped gases. The progressive microstructure evolution of the firn column with depth leads to pore closure and gas entrapment. Air transport models in the firn usually include a closed porosity profile based on available data. Pycnometry or melting-refreezing techniques have been used to obtain the ratio of closed to total porosity and air content in closed pores, respectively. X-ray computed tomography is complementary to these methods, as it enables to obtain the full pore network in 3D. This study takes advantage of this non-destructive technique to discuss the morphological evolution of pores on four different Antarctic sites. The computation of refined geometrical parameters for the very cold polar sites Dome C and Lock In (the two Antarctic plateau sites studied here) provides new information that could be used in further studies. The comparison of these two sites shows a more tortuous pore network at Lock In than at Dome C which should result in older gas ages in deep firn. A comprehensive estimation of the different errors related to X-ray tomography and to the sample variability has been performed. The procedure described here may be used as a guideline for further experimental characterization on firn samples. We show that the closed to total porosity ratio, which is classically used for the detection of pore closure, is strongly affected by the sample size, the image reconstruction and by spatial heterogeneities. In this work, we introduce an alternative parameter, the connectivity index, which is practically independent on sample size and image acquisition conditions, and that accurately predicts the close-off depth and density. Its strength also lies in its simple computation, without any assumption on the pore status (open or close). The close-off prediction is obtained for Dome C and Lock In, without any further numerical simulations on images (e.g. by permeability or diffusivity calculations).

## 1 Introduction

Ancient atmospheric air is embedded in polar ice caps, making them a major source of data for reconstructing past climates (Barnola et al., 1991; Battle et al., 1996). For example, the EPICA community traced climate variability back to 800 ky before present (Augustin et al., 2004) from ice cores drilled at Dome Concordia (also named Dome C). The firn layer (approximately the 50-100 top meters of the ice cores) is paramount in tackling the reconstruction of past climates, as the gradual densification of snow to ice leads to air entrapment. This firnication can last from a few centuries to a few thousand years before pore





occlusion with the consequence that the entrapped air is younger than the surrounding ice at a given depth (Schwander and Stauffer, 1984; Schwander et al., 1988). Linking unambiguously gas composition (from the air) and temperature evolution (from water isotopes in ice) in past climate conditions is challenging. In addition, the progressive closure of pores usually leads to a broad distribution of the trace gas ages. Seasonal layering of the polar firn also influences this closure (e.g. Stauffer et al.,

1985; Mitchell et al., 2015). In other words, the precise evaluation of the gas-ice age difference (Δage) requires a detailed understanding of the densification and gas trapping mechanisms.

A further complication is that, although the firn undergoes a similar densification with depth until it converts into solid ice, each polar site exhibits peculiar characteristics (local temperature, accumulation rate, impurity content, topography, etc...). Thus, climate proxies and air dating strongly depend on the polar sites investigated and there is now a substantial literature that

propose multi-sites studies (Landais et al., 2013; Veres et al., 2013; Bazin et al., 2013).

The firn column is classically divided into three distinct zones (Sowers et al., 1992). When considering air circulation, a convective zone is first encountered, where gases are mixed with the atmosphere in the top layers due to the high permeability of snow and/or the effect of wind. Deeper in the firn, in the diffusive zone, molecular diffusion dominates and gravitational fractionation also occurs. The lock-in depth (LID) is defined as the depth where gravitational fractionation stops (Battle et al.,

1996). In the deepest firn zone named lock-in zone (LIZ), the transport of gases in open pores becomes limited and eventually stops because pores are closing and are entrapping air. At the bottom of this zone, the close-off depth (COD) is the depth at which pores are fully isolated from the surface and where air cannot be pumped out of the firn anymore.

Successive densification mechanisms appear along the firn column. Snow undergoes grain rearrangement, fracture and sintering until a critical density of $550\,\mathrm{kg.m^{-3}}$ is reached (Anderson and Benson, 1963). This density corresponds to an approx-

imate porosity of 40%, close to the porosity that characterizes a Random Close Pack of spheres (Scott and Kilgour, 1969). As for many other granular materials, further densification requires the plastic deformation of the grains themselves. Above $550\,\mathrm{kg.m^{-3}}$, dislocation creep becomes the dominant mechanism for densification (Maeno and Ebinuma, 1983). The last stage of densification is porosity-related. When the density exceeds about $840\,\mathrm{kg.m^{-3}}$, pores are closed and further shrinking is driven by the difference between their internal pressure and the larger surrounding stress.

The mechanisms of pore closure in deep firn are of interest in order to understand the relationship between atmospheric signals and trace gas records in firn and ice. Models of gas transport in firn generally include a parameterization of the closed to total porosity ratio (e.g., Buizert et al., 2012). However, only a few closed porosity datasets are available (Stauffer et al., 1985; Schwander et al., 1993; Trudinger et al., 1997; Gregory et al., 2014; Schaller et al., 2017). Moreover, some recent studies emphasize the effect of firn layering, further motivating microstructural studies (e.g., Freitag et al., 2013; Gregory et al., 2014;

Mitchell et al., 2015; Rhodes et al., 2016; Fourteau et al., 2017).

The ratio of closed to total porosity may be obtained by two different techniques: pycnometry (e.g., Schwander and Stauffer, 1984), and X-ray computed tomography (e.g., Barnola et al., 2004). Air content measurements are generally performed by melting-refreezing techniques on samples taken well below the bubble closure zone and used as a proxy of the average air isolation density (e.g., Martinerie et al., 1992, 1994). Recently, Mitchell et al. (2015) measured air content in the LIZ of

the WAIS Divide ice core and used it as an indicator of the bubble closure rate. Each technique comes with its own issues.





Air content measurements, when performed on firn samples, may open artificially some closed pores that are weakly sealed and melting-refreezing is a destructive technique that forbids any further microstructural investigation (Raynaud et al., 1982). Pycnometry only provides a bulk closed porosity value without information on pore morphology. X-ray tomography scans a rather small volume of firn, thus questioning the representativeness of measured properties (Coléou et al., 2001). These three
techniques have all in common the issue of border effects. In particular, one has to make assumptions on the closed or open status of cut-pores. Regarding this issue, the effect of cut-pores seems nonetheless much smaller in the case of air content data obtained from bubbles well below the LIZ (e.g., Martinerie et al., 1992, 1994), than in the case of pycnometry or closed pores digitized by X-ray tomography on firn samples taken in the LIZ.

The X-ray tomography technique has received increasing interest in the last 20 years for investigating snow and firn mi-
crostructure (e.g., Flin et al., 2003; Barnola et al., 2004; Schneebeli and Sokratov, 2004; Freitag et al., 2004; Fujita et al., 2009; Lomonaco et al., 2011; Gregory et al., 2014). Focusing on the microstructural evolution that accompanies the closure process, Barnola et al. (2004) demonstrated the potential of the method by investigating the density and closed porosity evolution along the firn core of Vostok. They uncovered significant discrepancies between the ratio of closed porosity calculated from absorption images and measured by pycnometry. They attributed these differences to the small scanned volumes ($0.785\,\mathrm{cm}^3$)
compared to those used when performing pycnometry measurements ($100\,\mathrm{cm}^3$). Fujita et al. (2009) reported that the stratification of firn has an effect on several properties such as structural anisotropy, crystal orientated fabric or the total air content at Dome Fuji. Other Antarctic sites (WAIS Divide and the Megadunes region in East Antarctica) were also investigated. WAIS Divide exhibits high accumulation rate and relatively warm temperature, whereas Megadunes is very cold with a very low accumulation rate. Gregory et al. (2014) performed X-ray tomography and permeability measurements on these firn cores. They
concluded that the coldest site is characterized by a less tortuous pore morphology. Taking into account the strong layering exhibited in WAIS Divide, Gregory et al. (2014) also proposed that the air flow in polar firn is mainly controlled by pore structure and not so much by density variability. Focusing on the Summit site in Greenland, Lomonaco et al. (2011) studied closely fine-grained layers in the firn. Their results indicate that the ratio of the number of closed pores over the pore volume is marked by a sharp increase that related well with the LID. Gregory et al. (2014) also used such a parameter but no relation
with the LID can be observed on their Fig. 7. In contrast to this index, the increase of the closed to total porosity ratio is in better agreement with the LID. Overall, the literature on firn tomography concentrates on studying the evolution of structural parameters (such as permeability) and their relation to gas transport within the firn. It also focuses on differentiating polar sites, or relating microstructure to deformation mechanisms. However, despite giving useful information for densification modeling (e.g. Hörhold et al., 2011, 2012), these X-ray tomography studies do not provide an unambiguous prediction of LIZ and COD.
On that scope, the recent work of Schaller et al. (2017) depicts an abrupt closure of pores when reaching a critical porosity (around 10 %) that is independent of the temperature site. These authors studied three polar sites with a computed tomography allowing very large volumes of firn to be imaged.

More generally, for X-ray tomography to become a reliable reference technique for firn investigation, errors due to acquisition (voxel size or resolution), image analysis (image thresholding, labeling of pores) and sample variability (size and spatial
heterogeneities) and their impact on the microstructural properties should be thoroughly studied. This effort has already started



for modeling based on X-ray tomography images, whether for physical effective properties (Freitag et al., 2002; Courville et al., 2010) or for snow mechanics (Wautier et al., 2015; Rolland du Roscoat et al., 2007). For example, Freitag et al. (2002) and Courville et al. (2010) modeled the firn permeability by a lattice Boltzmann technique on a layered firn column using X-ray images. These authors worked out a representative volume element for permeability, going beyond the sole density as done by Coléou et al. (e.g., 2001).

In this context, here we investigate in detail the error sources that come with the process of X-ray tomography imaging on two East Antarctica firn cores originating from Dome C and Lock In (located 136 km away from the Concordia station towards Dumont d'Urville). We gather several microstructural parameters, such as density, closed to total porosity ratio, surface area or anisotropy, in order to study the closure of pores. We are able to discriminate which parameters correlate to the COD, as defined by the ultimate firn air pumping depth.

The paper is organized as follows. Section 2 details the characteristics of the sites studied here. Section 3 focuses on the errors originating from reconstruction and image processing of tomographic scans. In section 4, we investigate the scale effect and determine the errors generated on the main physical properties studied when decreasing the resolution in the tomographic scans. Section 5 focuses on the comparison between four different sites, with particular interest on the two sites studied in this work (namely Dome C and Lock-In), along with the polar sites detailed in Gregory et al. (2014). Section 6 takes advantage of image analysis tools to compute refined microstructural parameters that characterize the pore network throughout densification.

## 2 Site characteristics

Two sites in Antarctica are studied, Dome C and Lock In[1]. Dome C, near Concordia station, exhibits very cold mean annual temperature ($-55\,°C$) and low snow accumulation rate $\dot{b}$ ($25\,kg.m^{-2}.yr^{-1} \approx 2.5\,cm$ water equivalent per year, w.eq.yr$^{-1}$). Moreover, Dome C is at $3233\,m$ above sea level, with a thickness of nearly $3300\,m$ of ice. These characteristics make it an interesting site for drilling, as it shelters old ice (Augustin et al., 2004). Another distinctive feature of this site is the very narrow non-diffusive zone (lock-in zone, LIZ about 3 m Landais et al., 2006). The ultimate depth reached to sample air by pumping is $99.5\,m$ (Landais et al., 2006; Sturges et al., 2001). The ice cores studied come from the Volsol project carried out in 2010/2011 (Gautier et al., 2016). The firn samples originate from a unique ice core. Three ice samples (115.05, 123.34 and 132.07 m) from the DC12 ice core drilled in 2012 were also analyzed.

The Lock-In site is located 136 km away from the Concordia station towards the French base Dumont d'Urville. This site was sampled during the austral summer 2015-2016, and is characterized by a higher snow accumulation rate of about $\dot{b} = 4.5$ cm w.eq.yr$^{-1}$ (first guess from Verfaillie et al. (2012)). The borehole temperature is $-53.15\,°C$ at $20\,m$ depth. The LID is located between 96-100 m (Orsi, 2017), and air could not be pumped below approximately $108\,m$ indicating that the COD is deeper than for Dome C. A summary of the different site characteristics is given in Table 1.

---

[1] Note that Lock In is a polar site and does not refer to the lock-in phenomenon.





**Table 1.** Characteristics of different polar sites that were studied by X-ray tomography. $\dot{b}$ denotes the accumulation rate. The close-off depth is defined as the ultimate depth at which air could be sampled. The lock-in zone is the width between the LID (depth at which gravitational fractionation of $\delta^{15}N$ stops) and the COD and is defined as the LIZ width. The temperature is the mean annual temperature or borehole temperature.

| Site | Location | $\dot{b}$ (w.eq.yr$^{-1}$) | $\langle T \rangle$ (°C) | LIZ width (m) | COD (m) |
|---|---|---|---|---|---|
| Dome C [ab] | 75°6′ S, 123°21′ E | 2.5 cm | −55 | 3 | 99.5 |
| Lock In | 74°8.310′ S, 126°9.510′ E | ≈ 4.5 cm | −53.15 | $8 - 12^c$ | 108.3 |
| Dome Fuji[bd] | 77°19′ S, 39°40′ E | 2.1 cm | −57 | 0 | 104 |
| Vostok[abe] | 77°28′ S, 106°48′ E | 2.2 cm | −57 | 2 | 100 |
| WAIS Divide[fg] | 79°46.300′ S, 112°12.317′ W | 21 cm | −31 | ≈ 10 | 76.5 |
| Megadunes[fh] | 80°77.914′ S, 124°48.796′ E | < 4 cm | −49 | ≈ 4 | 68.5 |
| Summit[ij] | 72°34.48′ N, 37°38.24′ W | 21 cm | −31 | 10 | 80 |

[a] Bréant et al. (2017)
[b] Landais et al. (2006)
[c] Orsi (2017)
[d] Fujita et al. (2009)
[e] Barnola et al. (2004)
[f] Gregory et al. (2014)
[g] Battle et al. (2011)
[h] Severinghaus et al. (2010)
[i] Schwander et al. (1993)
[j] Witrant et al. (2012)

## 3 Methods

### 3.1 Acquisition parameters

X-ray micro-computed tomography was performed on samples coming from a large range of depths with a refined character-ization close to the depth at which closure of pores initiates. The majority of the sampling are samples labeled S (for small)

from Dome C (DC) and Lock In (LI), named DC-S12 and LI-S12 (see Table 2 for information on resolution of samples, which varies between 12 and 60 µm). S12 refers to small samples with voxel side length of 12 µm. These were drilled with a milling machine in slices of ice core such that the cylinder axis is along the core axis (vertical axis). The uncertainty of depth after drilling in slices is estimated to ± 0.05 m for all Lock In samples and most of Dome C samples (a few sample depths are known at ± 0.5 m). Samples were machined down to a diameter of 12 mm with a lathe. The machining operations were performed in

a cold room at −10 °C, usually the day before scanning. Scanning geometry was helical in order to image long samples (from 20 mm to 30 mm) with a volume more than twice the value typically studied by Barnola et al. (2004) and Gregory et al. (2014). Scans were performed at 60 kV with 800 radiographs over 4 turns leading to a scan time of approximately 25 minutes per sample. Samples were positioned in a cold cell, cooled by air at −10 °C thanks to the coupling of an air dryer and a cryostat. The temperature was controlled by a thermocouple positioned against the sample-holder inside the cell. Schematic of the set

up and geometry of the samples are shown in Figs. 1 and 2.



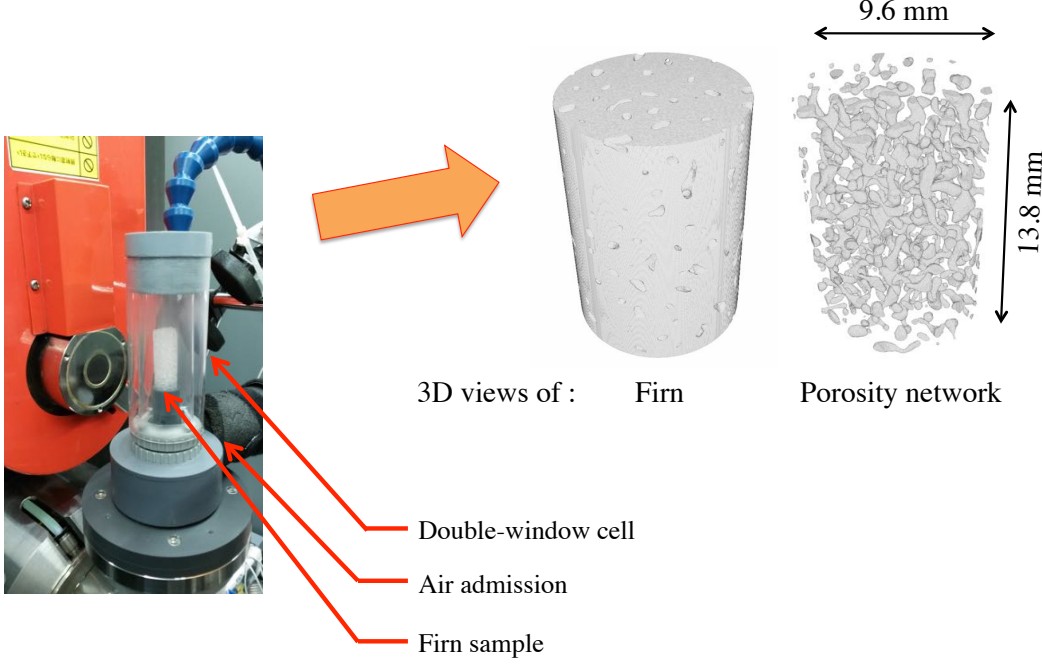

**Figure 1.** A firn sample from Dome C ($100.38$ m) inside the cold cell during X-ray tomography imaging, with associated 3D images of a firn sample and of its pore network.

Three samples labeled L (for Large) were also characterized to investigate the scale effect. They were placed in polystyrene boxes with an eutectic cold pack inside. Samples and cold pack inertia kept the temperature below $0\,°C$, ranging from $-18\,°C$ to an observed maximum at $-3\,°C$ after 1.5 hours (corresponding to two scans). According to metamorphism experiments of Kaempfer and Schneebeli (2007) and considering the thermal inertia of these large samples, any microstructural evolution should be negligible during the 1.5 hour scanning time. Moreover, two samples were scanned a first time, kept at $-10\,°C$ for 6 months and then scanned a second time. No evolution was observed. Volumes M (for Medium) were also locally scanned inside the L samples, i.e. a second scan is performed at higher resolution (using a voxel size of $30\,\mu m$ instead of $60\,\mu m$) and a larger acceleration voltage ($80\,kV$ instead of $60\,kV$). Sample characteristics are detailed in Table 2 and the extracted volumes are illustrated in Fig. 2.

## 3.2 Reconstruction and image processing

Reconstruction of the 3D sample structure is carried out with a filtered-back projection algorithm while image processing uses the free software Fiji (Schindelin et al., 2012). For S12 samples, a median filter of size 3 is first performed with the plugin





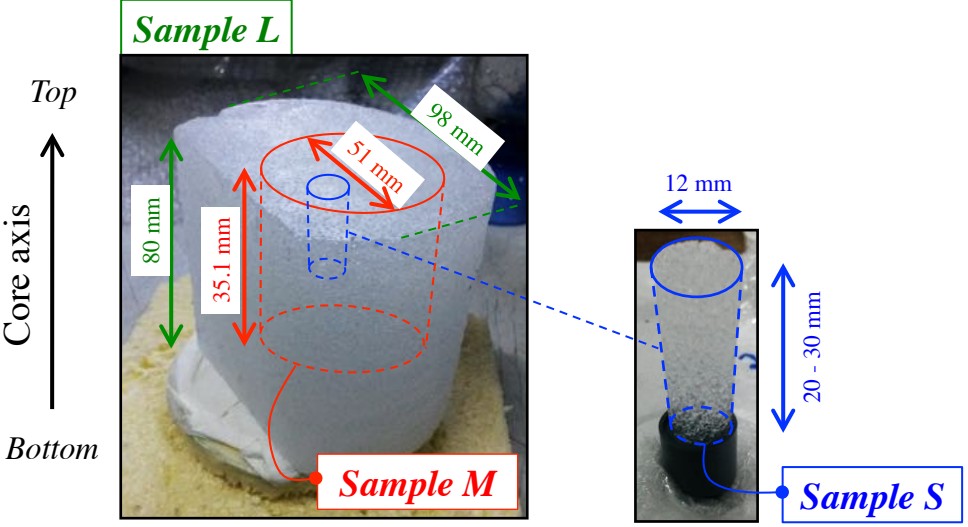

**Figure 2.** Schematic of the different sample sizes used in this work. Sample characteristics are listed in Table 2. In green, large samples L of the firn core were characterized by X-ray tomography using a voxel side length of 60 µm. Zooming inside the L samples enables to obtain medium sized samples (typical of the size used in pycnometry measurements) named M and shown in red. These were scanned at 30 µm (DC-M30). More than 40 samples (shown in blue) of size S were machined separately and scanned at 12 µm for both sites (samples DC-S12 and LI-S12).

**Table 2.** 3D image characteristics from Dome C (DC) and Lock In (LI). Samples are scanned with an acceleration voltage of 60 kV except for DC-M30 and DC-S30 volumes, scanned under 80 kV. DC-L60 samples have a polyhedron geometry. M and S volumes are cylindrical. DC-C12 and LI-C12 are cubic (C) subvolumes of DC-S12 and LI-S12, respectively. For DC-L60 to DC-S30, the Region Of Interest (ROI) comes from the same 3 large samples (L). DC-S12 and DC-C12 come from the same 29 S samples . LI-S12 and LI-C12 come from the same 13 S samples . The number at the end of volume names refers to the voxel size in microns.

| Origin | Name | Diameter / Side (mm) of ROI | ROI volume (cm³) | Voxel size (µm) | Number of ROI volumes | Depth (m) | Remarks |
|---|---|---|---|---|---|---|---|
| DC | DC-L60 | | > 400 | 60 | 3 | 87.34 ; 94.5 ; 100.33 | Slice of ice core from Dome C |
| DC | DC-M60 | 51 | ≈ 72 | 60 | 3 | " | Zoom within the DC-L60 samples |
| DC | DC-M30 | 51 | ≈ 72 | 30 | 3 | " | Scans within the L samples |
| DC | DC-S30 | 9.6 | 1.305 | 30 | 39 | " | Sub-volumes from the DC-M30 samples |
| DC | DC-S12 | 9.6 | 0.782 - 1.781 | 12 | 29 | 22.33 - 100.38 | Cylinders taken from ice core slices |
| DC | DC-C12 | 6.72 | 0.303 | 12 | 29 | " | Cubic sub-volumes from the DC-S12 samples |
| LI | LI-S12 | 9.6 | 1.303 - 1.738 | 12 | 13 | 66 - 120 | Samples from Lock In |
| LI | LI-C12 | 6.72 | 0.303 | 12 | 13 | " | Cubic sub-volumes from the LI-S12 samples |

Analysis 3D developed by Boulos et al. (2013) in order to reduce noise and enhance contrast. The high contrast between air and ice ensures a straightforward thresholding of 3D binary images. Outliers (persisting noise artefacts) of 2 or 3 voxels were removed, leading to the loss of some micro-pores with diameter smaller than 36 µm. This represents less than 0.7% of the total porosity and leads to an estimated 0.1% error on density.

5  A cylindrical region of interest (ROI) is systematically extracted from the digitized sample, to ensure that the analysis is not perturbed by ice fragments localized on the sample borders (typically originating from machining). Pore labeling is also





performed on this ROI by the Analysis 3D plugin (Boulos et al., 2013). All extracted ROI's are detailed in Table 2. The errors introduced successively by the potential sublimation of matter inside the cell during scanning time, the reconstruction, and the image processing (filtering and thresholding), were determined as follows. On a smaller sample height ($h =12\,\mathrm{mm}$), tomographic scans lasting 5 minutes were repeated during three hours on a $86\,\mathrm{m}$ deep sample from Dome C, first, every five

minutes, then with increasing intervals of time. The results on these 12 scans have shown that sublimation was limited to the outward $0.2\,\mathrm{mm}$ of the cylindrical sample after 30 minutes, even for a porosity for which percolation is significant. As borders are eliminated after extraction of the ROI (diameter $9.6\,\mathrm{mm}$ for samples S of diameter $12\,\mathrm{mm}$), there is no influence of the dry air circulation around the firn in the cold cell. Multiple acquisitions of the same sample allows the errors combining acquisition, reconstruction and Fiji processing (median filter + thresholding + outliers removal) for such a scan, to be computed.

Several firn properties are determined in this work such as the density, the specific surface area (SSA) or other pore related parameters. The SSA is defined as the surface of the pores over the total sample volume. The surface is computed with the marching cube algorithm (Lorensen and Cline, 1987). The SSA is used to quantify the air-ice interactions (e.g., to compute the snow-to-atmosphere flux of adsorbed molecules on snow (Domine et al., 2008)) or to work out the grain size of firn cores (Linow et al., 2012). The volume used to compute total porosity is calculated by counting voxels. The closed to total porosity

ratio and the connectivity index (the ratio of the volume of the largest pore to the total pore volume) are also systematically calculated. We characterize a closed pore as a pore that does not touch the border of the ROI at the sample resolution. The pore volume fraction is defined as the closed to total porosity ratio (in per cent). Results for these parameters are discussed in section 4.

The 12 successive scans of the same sample are used to compute standard deviations. At $86\,\mathrm{m}$ depth, absolute errors are

$1.2\,\mathrm{kg.m}^{-3}$ on density, $0.008\,\mathrm{mm}^{-1}$ on the specific surface area (SSA), $0.068\,\%$ for the connectivity index and $0.25\,\%$ on the closed porosity ratio, while relative standard errors are respectively $0.15\,\%$, $0.81\,\%$, $0.11\,\%$ and $3.1\,\%$. Since this procedure was performed on only one sample, we assume that the relative errors associated to each microstructural parameter are the same for all depths. In other words, we assume that the precision (from X-ray scans and image processing) on a given property is independent of depth for all samples S12.

## 4   Representativeness of morphological parameters

Conducting a refined X-ray characterization and comparison of polar firn requires a correct estimation of possible errors. Besides precision, several other sources of errors exist which do not have the same influence on the results. These errors can come from spatial heterogeneity, size of the ROI, image resolution, image processing and labeling assumptions.

This section focuses on giving estimates of these errors on the important parameters studied: the density, the closed porosity

ratio, the connectivity index and the specific surface area. These errors depend on depth, except for the processing ones for which we assume an independent relative value. Results presented hereafter can be useful for future tomographic-based studies that should advantageously include error and variability estimations.





## 4.1 Density

Densities were determined directly from binary images using $\rho_{ice} = 917$ kg.m$^{-3}$ and are shown against depth for both sites in Fig. 3a. We also attempted to measure density by weighting samples as done in Gregory et al. (2014). However, we observed that this method led to a very large dispersion, which may be linked to a too small sample volume. In any case, we chose to link the density measurement directly to the local scanned volume. We believe that this is preferable than to ascribe a density originating from a much larger volume as carried out by Gregory et al. (2014), which may not be representative of the actual scanned sample.

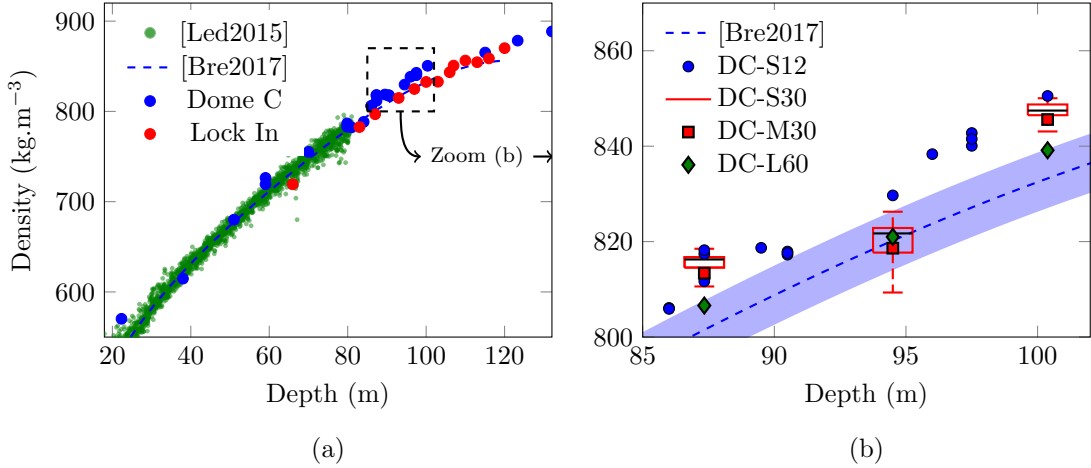

(a)  (b)

**Figure 3.** (a) Evolution of density with depth for Dome C and Lock In. (b) Zoom-in from (a) focused on Dome C for various ROI sizes (S, M, L) and voxel sizes (12, 30 and 60 μm). Boxplots contain data from DC-S30 volumes. Hereafter, boxes represent lower and upper quartiles, the black lines inside being the median values. The confidence interval is set to account for all the data (no outliers). Green dots [Led2015] on (a) are data obtained by Leduc-Leballeur et al. (2015) from volume and mass measurements of 5 cm thick samples from a Dome C ice core down to 80 m. The blue dashed line [Bre2017] corresponds to a mean density profile of Dome C averaged on a 25 cm scale from Bréant et al. (2017). The blue shaded area on (b) represents a standard deviation of 6.2 kg.m$^{-3}$ between the fit and density data as given by Bréant et al. (2017).

Fig. 3a illustrates the increase in density with depth. It also shows that for a given depth, the density at Dome C is larger than at Lock In. As stated above, the relative error in measuring density is less than 0.15 % and thus is not shown here. For approximately the same depth, differences between the density of Dome C and Lock In samples are between 1.8 and 2.6 % (in the range of depths studied).

The density of Dome C samples obtained by X-ray tomography superimposes correctly with the scattered data of Leduc-Leballeur et al. (2015) that worked out density by measuring the mass and volume of samples 5 cm long. The high resolution density measurements performed by Leduc-Leballeur et al. (2015) required a large number of samples (30 to 37 samples by 2-meter layers), thus explaining the observed scattering in their measurements.

Densification of firn into ice leads to a transition where an open pore, that enabled air to flow, begins to separate in different pores. This closure of pores occurs all along the firn core, but is more pronounced just before the close-off, typically when





density ranges between $800$ and $840\,\mathrm{kg.m^{-3}}$ where we gathered more data points. Fig. 3b is a zoom-in on this region for Dome C. A mean density profile was proposed at Dome C (Bréant et al., 2017) and goes below $80\,\mathrm{m}$, with standard deviation of $6.2\,\mathrm{kg.m^{-3}}$. While DC-S12 densities and the density profile [Bre2017] seems in accordance before $85\,\mathrm{m}$ in Fig. 3a, they diverge deeper in the firn in Fig. 3b, with DC-S12 densities being above the density profile and its standard deviation. Except for

samples from $94.5\,\mathrm{m}$, calculating the density from a binary image seems to overestimate the mean density by approximately $2\,\%$ in the 85-100 m range. Three DC-S12 samples were extracted from the same slice of ice core at $86\,\mathrm{m}$, $91\,\mathrm{m}$ and $98\,\mathrm{m}$ depth. These triplets of points are barely distinguishable. The red boxplots shown in this figure represent the extrema on 13 sub-volume measurements (small DC-S30 images) originating from a medium-sized DC-M30 sample. These boxplots show dispersion inside medium-sized samples (sub-volumes of $1.3\,\mathrm{cm^3}$ inside $72\,\mathrm{cm^3}$). The difference between the maximal and

minimal values for samples from the same depth is less than $2.5\,\%$ (maximum variability is obtained at $94.5\,\mathrm{m}$). Thus, the effects of spatial heterogeneities and of the size of the ROI are limited but are the main contributions to the variability of the density. Indeed, three additional samples S were scanned at a voxel side length of 12 and $30\,\mathrm{\mu m}$ and exhibited differences of less than $0.1\,\%$ for density. Similarly, the difference between densities of the samples DC-M30 and DC-M60 is less than $1\,\%$ (see Table 3). In conclusion, the density is known quite precisely, and DC-S12 samples are large enough to be considered

representative in terms of the density.

## 4.2   Closed porosity

The closed to total porosity ratio (CP) is obtained by dividing the total volume of closed pores in the ROI by the total volume of pores. Fig. 4 shows the evolution of closed pores (in red) within the firn pore network for Dome C and Lock In. The number of closed-pores increases with depth, and separation persists even for pores already closed. We observed that the separation of

pores arises by pinching of the channel linking two larger globular portions of the pore as illustrated by Fig. 4i. Areas A and B in Fig. 4i show examples of channels on the verge to disappear, while areas C and D highlight newly pinched pores (dead ends).





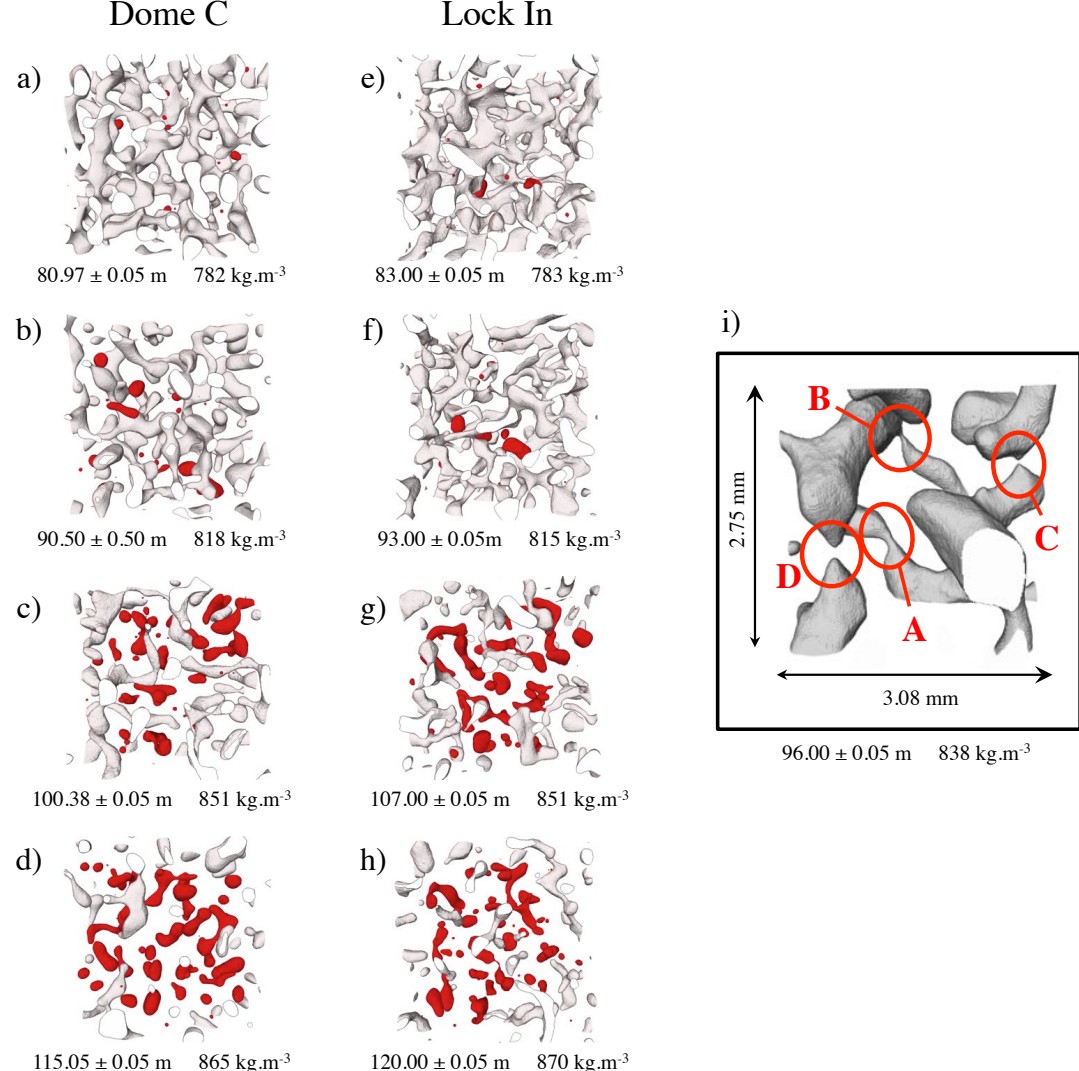

**Figure 4.** Evolution of closed pores (red) within the firn porosity network ( grey) for different depths at Dome C and Lock In sites for DC-C12 and LI-C12 volumes (cross-section dimension is 6.72x6.72 mm$^2$). Note that (a-e), (b-f), (c-g) and (d-h) have approximately the same density. (i) Close-up view from a DC-C12 volume of a few channels between pores at different stages of the pinching process illustrated by areas A-B-C-D.

Fig. 5a shows a steep increase of the closed porosity ratio (CP) for DC-S12 samples, starting at $80$ m depth (corresponding to approximately $780$ kg.m$^{-3}$). As in Fig. 3b, at $86$ m, $91$ m and $98$ m depths, three DC-S12 samples were extracted from the same slice of ice core to evaluate dispersion (area A in Fig. 5a). The DC-S30 volumes are represented by boxplots, which are rather large (especially at $94.5$ m depth). Boxes represent lower and upper quartiles of the distribution of the closed porosity ratio, with median value in black. The confidence intervals take into account all data points. Spatial heterogeneities on the closed



porosity ratio strongly depend on depth. This variability between small volumes is particularly pronounced at 94.5 and 98 m, but smaller when closed pores are few (at 87.34 m) or after the COD as defined by the last firn air pumping depth (sample from 100.33 m, whereas the COD is at 99.5 m (Landais et al., 2006; Witrant et al., 2012)). This leaves two possibilities. First, the firn may be very heterogeneous horizontally, meaning that a single DC-S12 sample is not representative enough for the closed

porosity ratio of the firn for this particular depth. Second, it could mean that our definition of a closed pore is not appropriate, due to the sample boundary conditions. Note that for such depths (94.5 and 98 m), the mean size of a closed pore is still likely too large to hold inside the DC-S12 samples.

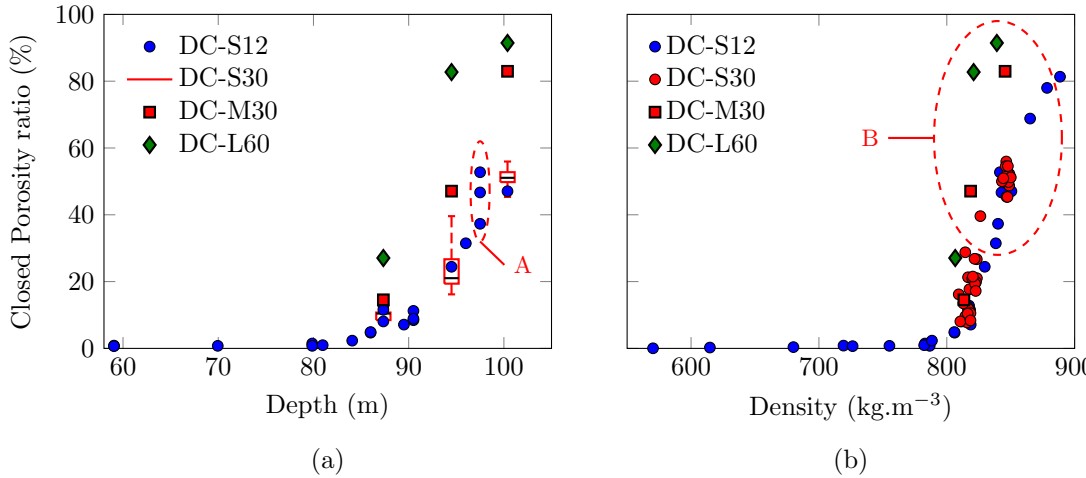

**Figure 5.** Evolution of the closed porosity (CP) in percentage of total porosity against (a) depth and (b) density for various sample sizes (S, M, L) and voxel sizes (12, 30 and 60 μm). The variability of the closed porosity of various samples from the same depth are represented by red boxplots on (a) and red dots on (b) as densities are not exactly identical.

     ROI size also has a strong influence on the estimation of the closed porosity ratio. Indeed, the larger the volume, the larger the closed porosity ratio (Fig. 5). Note that this effect is more pronounced at 100.33 m (compare volumes DC-M30 and DC-

S12) than at 87.34 m depth. Errors due to coarsening in resolution are summarized in Table 3 and reveal a significant effect on the results between the DC-M30 and DC-M60 volumes[2]. The closed porosity ratio is always larger when increasing the voxel size, and this effect is similar to the one of the ROI volume. For labeling, the type of connectivity (6 or 26) is also investigated, and its influence on the results is limited for all resolutions considered. Using 26 connected voxels does enhance the possibility for different pores to be more connected that it is with only 6 connected voxels as cubic voxels are not only

connected through faces (6) but also through edges and corners (26). It is especially true at the depths 87.34 m and 94.5 m as this is where pinching is very pronounced and on the verge to cut pore channels. Thus a change in the definition of the pores as implied by the use of 26 connected voxels might connect some closed pores together or to the surface. Similarly, coarsening the resolution means a different threshold value when segmenting the image, thus a possible separation of pores or channels. This seems to be especially true for low resolution, e.g. here for a voxel size of 60 μm. Indeed, differences related to voxel size

---

[2]   DC-M60 and DC-M30 have the same ROI, but are scanned for a voxel side length of 30 μm and 60 μm respectively (see Table 2)





between the same volumes DC-S12 and DC-S30 are smaller, as the relative error for the closed porosity ratio is less than $7\%$ with respect to the $12\,\mu\mathrm{m}$ resolution (see Table 4). This is relatively low compared to the extent of the box plot represented in Fig. 5a, which spans from $16\%$ to $40\%$ at $94.5\,\mathrm{m}$ depth.

All the samples and the sub-volumes extracted are plotted versus density on Fig. 5b. While the smallest volumes seem to
follow a trend, the largest ones are still spread (area B in Fig. 5b). From this plot it is impossible to determine at which density or closed porosity ratio, the close-off could be unambiguously identified. Considering air content in firn, the parameterization from Goujon et al. (2003) assumes that the volume fraction of closed pores is approximately $37\%$ at the average air isolation level (defined as the average porous volume or density of air isolation calculated from air content measurements (Martinerie et al., 1992, 1994)). On the other hand, at $94.5\,\mathrm{m}$ depth DC-M30 and DC-L60 volumes showed that $47\%$ to $82\%$ of pores
are closed at $94.5\,\mathrm{m}$ depth for example. Such a high values could suggest that the close-off occurs at shallower depth than commonly accepted. However the density of this sample ($\rho = 821\ \mathrm{kg.m}^{-3}$) is lower than the average air isolation density obtained by Martinerie et al. (1992) ($\rho = 840\ \mathrm{kg.m}^{-3}$), and air was pumped out of the firn down to $99.5\,\mathrm{m}$ (Landais et al., 2006), which is five meters below. Additionally, the rather large voxel size leads to separation of pores that are linked by channels whose diameter are below the image resolution, increasing artificially the closed porosity ratio.

Therefore, the rather large errors on the closed porosity ratio of DC-L60 samples calls for caution in interpreting results (see Table 3). In short, the most appropriate use of the closed porosity ratio requires large volumes associated with high resolutions.

### 4.3 Connectivity Index

In this section, we propose an alternative indicator of the pore closure, which is much less sensitive to the source of errors that characterize the closed porosity ratio, especially the sample size. The connectivity index (CI) - volume of the largest pore
divided by the total volume of pores - was introduced by Babin et al. (2006) to depict the void coalescence in bread. It proved useful (under the name of "interlinkage parameter") to quantify the coalescence of cavities during superplastic deformation of an aluminum alloy (Martin et al., 2000), or as a criterion to optimize box sizes when smoothing density maps of graphite with inclusions (Babout et al., 2006). This index describes, as the closed porosity ratio does, the evolution of closed pores. However, it is not dependent on sample boundary conditions, since all pores are considered. This is critical for small volumes as the ratio
of surface over volume becomes significant. One major setback is that it is very dependent upon the volume of pores.





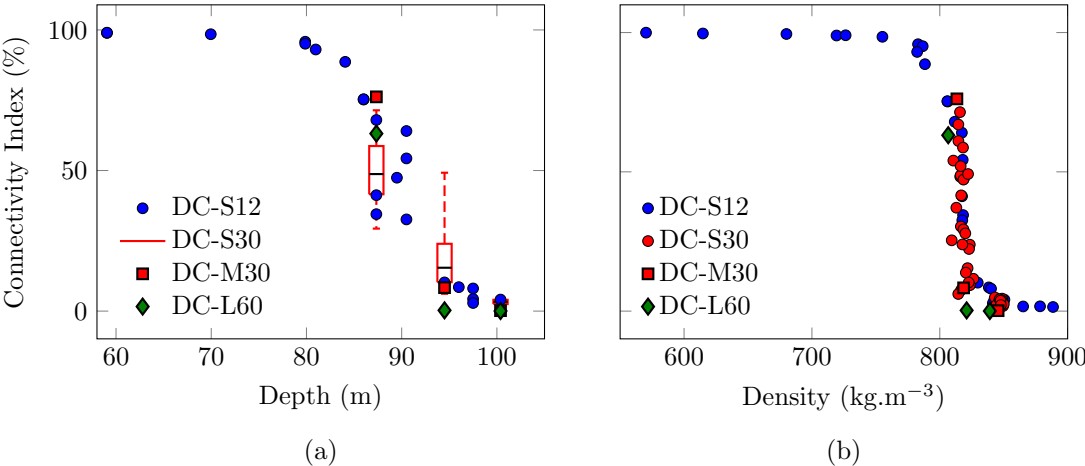

**Figure 6.** Evolution of the connectivity index (volume of the largest pore divided by total pore volume) with (a) depth and (b) density. The variability of the connectivity index of various samples from the same depth are represented by red boxplots on (a) and red dots on (b) as the densities are not exactly identical.

As shown in Fig. 6, the connectivity index is maximum and close to $100\,\%$ when the porosity is totally open (one large pore overcomes all others) (the pore network is almost fully interconnected), while it is very small (less than $0.2\,\%$ , see Table 3) when all pores are closed. The connectivity index drops around $780\,\mathrm{kg.m^{-3}}$ ($\approx 80\,\mathrm{m}$), and stagnates after $850\,\mathrm{kg.m^{-3}}$ ($100\,\mathrm{m}$). Sub-sampling leads to large boxplots thus large variability at $87.34$ and $94.5\,\mathrm{m}$, while it is much smaller at $100.33\,\mathrm{m}$. Thus

spatial heterogeneity is pronounced during the drop of the connectivity index. DC-M30 or DC-L60 volumes are inside or close to the boxplot. This is in contrast with the closed porosity ratio in Fig. 5 where DC-M30 or DC-L60 volumes are far from the boxplot. This means that the use of the connectivity index induces a much weaker effect of the ROI volume. Comparing resolution and type of connectivity of the DC-S12 and DC-S30 volumes, results for the connectivity index are similar to the closed porosity ratio (differences of less than $7\,\%$ in Table 4). Table 3 also lists errors between the DC-M30 and DC-M60

volumes for the connectivity index. Overall, these ROIs fall in the box plots of the DC-S30 volumes (see Fig. 6a) .

In conclusion, the variability of the connectivity index comes mostly from the horizontal spatial heterogeneity. It is much less sensitive to the volume of the samples, the resolution or the type of connected voxels than the closed porosity ratio. Interestingly, Fig. 6b indicates that the connectivity index points fall on a master curve when plotted against the density. In particular, a clear drop of the connectivity index is observed at a density of about $830\,\mathrm{kg.m^{-3}}$. Above this density, DC-L60

and DC-M30 exhibit no large pores anymore. At Dome C, the last firn air pumping depth is $99.5\,\mathrm{m}$ (Landais et al., 2006) and Martinerie et al. (1992) obtained a density of $840\,\mathrm{kg.m^{-3}}$ at average air isolation level from air content measurements. Thus, according to Fig. 6, the connectivity index is a good predictor of the close-off. However, it does not give any information on the LID, which is required for $\Delta$age estimation.





## 4.4 Specific surface area

The SSA involves the quantification of the surface of the pores. This measurement is voxel size dependent due to the surface roughness. However, as shown by Tables 3-5, differences of voxel size do not lead to too large an error. Changes due to ROI volume and connectivities are also not significant for the SSA. These results are very similar to those obtained for density, however the effect of the spatial heterogeneity is more pronounced. Indeed, in the worst case scenario, relative differences for the same depth can reach up to $20\,\%$. All the other sources of errors give results in the range of the variability of the horizontal spatial heterogeneities. Also, all data points are known very precisely as pores are defined by a large amount of voxels. In the following, the surface-area-to-volume ratio is used instead of the SSA for comparison with Gregory et al. (2014). Both parameters mostly reflect the pore surface. However the volume considered for the surface-area-to-volume ratio is the porous phase instead of the whole ROI. The relative standard error for the surface-area-to-volume ratio ($S/V$) is $0.47\,\%$ (Table 5).

**Table 3.** Results and differences between DC-M30 and M60 samples. Results with 6 and 26 connected voxels are shown.

| Depth (m) | | 87.34 | | | 94.50 | | | 100.33 | |
|---|---|---|---|---|---|---|---|---|---|
| Vox. size (µm) | 30 | 60 | Δ | 30 | 60 | Δ | 30 | 60 | Δ |
| $\rho$ (kg.m$^{-3}$) | 813.4 | 807.6 | -5.8 | 818.6 | 818.8 | +0.2 | 845.6 | 839.7 | -5.9 |
| SSA (mm$^{-1}$) | 0.970 | 0.954 | -0.016 | 1.015 | 0.973 | -0.042 | 0.711 | 0.713 | -0.002 |
| CP (%) : co-6 | 14.55 | 23.23 | +8.68 | 47.08 | 69.06 | +21.98 | 82.96 | 83.62 | +0.66 |
| CP (%) : co-26 | 14.41 | 21.65 | +7.24 | 46.58 | 65.67 | +19.09 | 82.87 | 83.22 | +0.35 |
| IC (%) : co-6 | 76.20 | 60.37 | -15.83 | 8.35 | 1.46 | -6.89 | 0.16 | 0.17 | +0.01 |
| IC (%) : co-26 | 76.60 | 62.82 | -13.78 | 14.74 | 2.08 | -12.66 | 0.16 | 0.17 | +0.01 |

**Table 4.** Results and differences between DC-S12 and S30 samples.

| Depth (m) | | 89.5 | | | 94.5 | |
|---|---|---|---|---|---|---|
| Vox. size (µm) | 12 | 30 | Δ | 12 | 30 | Δ |
| CP (%) | 12.14 | 13.06 | +0.92 | 24.64 | 24.41 | -0.25 |
| IC (%) | 34.38 | 33.82 | -0.56 | 10.31 | 10.32 | +0.01 |

**Table 5.** Absolute and relative errors for two DC-S12 samples. Relative errors are determined thanks to Section 3.2.

| Depth (m) | 84.07 | Abs. Err. | 97.5 | Abs. Err. | Rel. Err. |
|---|---|---|---|---|---|
| $\rho$ (kg.m$^{-3}$) | 788.4 | ±1.2 | 841.5 | ±1.3 | 0.15 % |
| SSA (mm$^{-1}$) | 1.192 | ±0.010 | 0.750 | ±0.006 | 0.81 % |
| $S/V$ (mm$^{-1}$) | 8.478 | ±0.040 | 9.108 | ±0.043 | 0.47 % |
| CP (%) : co-6 | 2.32 | ±0.07 | 52.74 | ±1.63 | 3.1 % |
| CP (%) : co-26 | 2.32 | ±0.07 | 52.03 | ±1.61 | 3.1 % |
| IC (%) : co-6 | 88.654 | ±0.098 | 2.883 | ±0.003 | 0.11 % |
| IC (%) : co-26 | 88.654 | ±0.098 | 2.883 | ±0.003 | 0.11 % |





### 4.5 Key results on error estimations

To sum up, in this section we have characterized separately the errors associated with the calculated microstructural parameters when choosing the resolution, the type of connected voxels and the size of the sample. Table 3 and Figs. 5 and 6 showed that the variability of the results can be significant and is dependent on the depth of the sample. The closed porosity ratio was

shown to be dependent on ROI size, voxel size and spatial heterogeneities. When plotted versus density, the connectivity index reveals to be a very appropriate parameter to describe the progressive pore closure. The proposed connectivity index is a more discriminant parameter to describe the pore closure, as it is foremost dependent on spatial heterogeneities.

    The precise determination of microstructural parameters requires a small voxel size. With X-ray tomography, this means a small sample volume. Therefore, in the following, only samples whose voxel side length is 12 µm are studied and compared.

In order to avoid to overload the following figures, uncertainties are not displayed but Table 5 allows the absolute uncertainties for two distinct depths of the DC-S12 samples to be reported.

## 5   Multi-site comparisons

In this section, we compare our data retrieved from Dome C and Lock In to those originating from WAIS Divide (West Antarctica) and Megadunes (East Antarctica) from Gregory et al. (2014), which were also analyzed with X-ray tomography.

We focus on the closed porosity ratio and morphological parameters that can be compared for all four sites. We note however that the depth intervals probed for WAIS Divide and Megadunes sites are smaller than ours. Temperature and accumulation conditions for WAIS Divide and Megadunes are listed in Table 1, porosity is closing at a much shallower depth, between 60 and 80 m for WAIS Divide and Megadunes, as indicated by Fig. 7a, than Dome C and Lock In. In contrast, pore closure occurs below 80 m for Dome C and Lock In. The Lock In site has a larger accumulation rate and is a bit warmer than Dome C

but exhibits a deeper closure of pores. When plotting the percentage of closed porosity against density, Fig. 7b shows that all points fall approximately on a master curve. This supports the idea that the close-off arises on first approximation at a particular density. Note that the Megadunes site is peculiar, as a dune experiences continuous snow deposition on its sides, and has an average of zero accumulation in "hiatus" zones (Courville et al., 2007; Severinghaus et al., 2010).



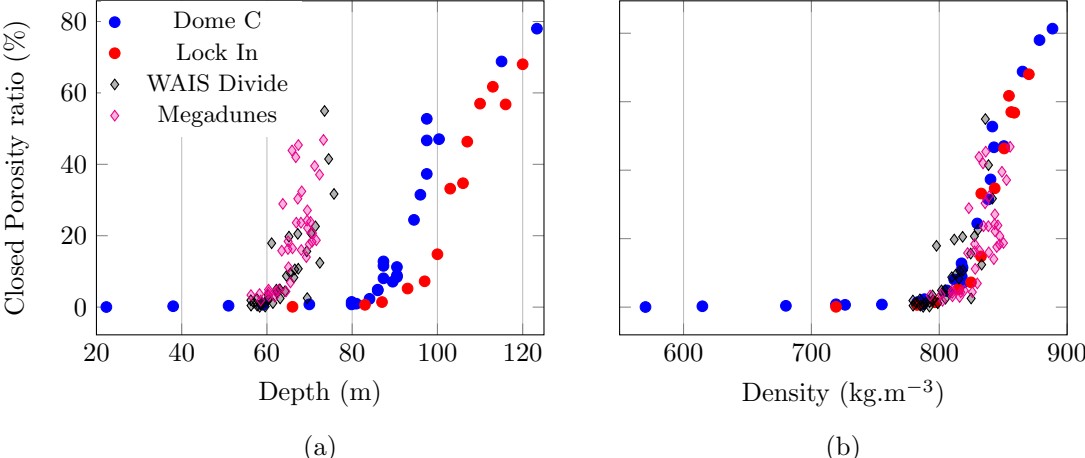

**Figure 7.** Comparison between four polar sites - Dome C, Lock In, WAIS Divide and Megadunes - of the closed porosity evolution with (a) depth and (b) density. Diamonds are data points from Gregory et al. (2014).

Figure 8 shows three morphological parameters of interest (degree of anisotropy, surface-area-to-volume ratio of pores, and structure model index (SMI)) that can be compared for all four sites. The evolution of these parameters is shown against depth (Figs. 8a,c,e) and against density (Figs. 8b,d,f). These geometrical parameters provide valuable information on the pore network morphology that could be useful for the understanding of firnification.

Figures 8a-b show the degree of anisotropy that was determined using the BoneJ plugin (Doube et al., 2010) on the pore phase. This parameter is calculated using a 3D mean intercept length method that works out directly all possible directions. An ellipsoid is fitted to the scaled intercepted points, giving eigenvalues for the lengths of the ellipsoid axes. The degree of anisotropy of the morphology is 0 for a fully isotropic structure and tends to unity when all objects are aligned along a direction. Figs. 8a-b indicate that for all four sites, the degree of anisotropy is rather low ($\leq 0.4$), with a mean value of the order of 0.2.

No clear trend is observable with depth or density.

The surface-area-to-volume ratio ($S/V$) of the pores is depicted in Figs. 8c-d. $S/V$ may be seen as a good descriptor of the complexity of the pore network morphology. In the case of pores as shown in Fig. 4, pore roughness is similar for both sites (pore surfaces are smooth). Therefore, a less tortuous network leads to a smaller $S/V$ value. In contrast with anisotropy (Fig. 8a-b), a clear difference is observable between Dome C and Lock In sites on one hand, and WAIS Divide and Megadune sites (Gregory et al., 2014) on the other hand. Our data indicate a fairly smooth trend for Dome C and Lock In sites, whereas WAIS Divide Megadunes

sites (Gregory et al., 2014) show much more variability on a rather limited depth interval. Gregory et al. (2014) characterized samples both from fine-grained and coarse-grained layers. The microstructural differences that inherently come with fine and coarse grains should explain the variability observed in their results. Concentrating on the $S/V$ ratio against density (Fig. 8d), and focusing on the limited common density interval (750 - 850 kg.m$^{-3}$), we note that WAIS Divide exhibits the most tortuous

pore network. Megadunes leads to a pore network which $S/V$ values are in between those of Dome C and Lock In, but again with a much larger variability. Although relatively limited, there is a clear difference between Dome C and Lock In, with Dome





C consistently exhibiting smaller $S/V$ values than Lock In, either for the same density or for the same depth. Thus, Lock In presents, for a given depth or density, a more tortuous pore network than Dome C. Dome C and Lock In S/V values follow approximately a linear increase with depth.

The Structure Model Index (SMI) is a popular index in the bone research community which is implemented in the SkyScan software CT-analyzer (Sky, 2017) used by Gregory et al. (2014). It was also used to analyze snow metamorphism for instance (Schneebeli and Sokratov, 2004; Kaempfer and Schneebeli, 2007). The SMI describes the shape of pores, with positive values for convex shapes and negative values for concave ones. The SMI tends to 0 for plates, 3 for rods and 4 for spherical particles. Two different methods are available to compute the SMI: the Skyscan plugin and the BoneJ plugin with the Hildebrand algorithms (Hildebrand and Rüegsegger, 1997). As highlighted by Salmon et al. (2015), the Skyscan plugin consistently leads to smaller SMI values than the Hildebrand plugin. For sake of comparison with the data of Gregory et al. (2014), Figs. 8e-f show the Structure Model Index (SMI) of the porous phase obtained with both methods. As for the SSA, the SMI is computed using the pore surface area. Consequently, uncertainties are negligible and not shown.

As for the $S/V$ parameter, the SMI computed by Gregory et al. (2014) (using the Skyscan plugin) exhibits a much larger variability than ours. In other words, the pore shapes measured at WAIS Divide range from semi-cylindrical to nearly spherical for a rather limited span of depth (55 - 70 m) or density. In contrast, data points from Dome C and Lock In follow a clear trend, which suggests a simple evolution from rod-like to sphere-like shape for increasing depth or density.

We have checked that the difference between WAIS Divide and Megadunes on one hand, and Dome C and Lock In data on the other hand does not originate from the different methodologies used in the two groups by computing the SMI of Dome C and Lock In with the same plugin used for WAIS Divide and Megadunes. As shown by Figs. 8e-f, the Skyscan plugin leads indeed to smaller SMI values than Hildebrand. But this does not contradict the general picture that Dome C - Lock In and WAIS Divide - Megadunes sites differ markedly when considering the morphology of pores. In any case, we believe that the linear evolution observed in Figs. 8e-f for Dome C and Lock In is pertinent as it relates well with the change in pore morphology shown in Fig. 4.





**Figure 8.** Anisotropy (a-b), surface-area-to-volume ratio (c-d) and Structure Model Index (e-f) plotted against depth and density for Dome C and Lock In (this work) as well as for WAIS Divide and Megadunes (Gregory et al., 2014). The SMI index for Dome C and Lock In is computed using both the Hildebrand method (Hildebrand and Rüegsegger, 1997) and the Skyscan plugin.





## 6 Refined geometrical parameters for Dome C and Lock In sites

In the preceding section, we have compared four different sites using available data from the literature. In this section, we take advantage of the collected 3D images for Dome C and Lock In to compute refined geometrical parameters. They should bring new insights into the intimate structure of firn and its evolution, and are also useful properties that can be used for diffusion or permeability modeling.

We have chosen to plot these parameters as a function of density alone but their evolution against depth can be obtained by fitting the data points in Fig. 3. Figure 9 sums up and defines these parameters: medium chord length (Fig. 9a), Chamber over throat size ratio (Fig. 9b), maximal path diameter (Fig. 9c) and connectivity index defined in section 4.3 (Fig. 9d). The parameters in Figs. 9a-c have been obtained by using the commercial software Geodict (Geo, 2014). Definitions of these parameters are given hereafter. Volumes DC-C12 and LI-C12 were used, as the software requires parallelepipedic ROIs.

Figure 9a shows the medium chord length in all three directions. The corresponding sketch only illustrates the intercepts in the $z$ direction. The intercept lengths are measured and averaged for each direction. For each density, the medium chord length can be considered isotropic because no preferential direction can be determined. Both DC-C12 and LI-C12 depict a linear decrease of about $50\,\%$ from $22.33\,\mathrm{m}$ to $100.38\,\mathrm{m}$ depth for Dome C. For a given density, the medium chord length of LI-C12 is always smaller than that of the DC-C12 volumes. This confirms the more complex shape of pore morphology for Lock In as observed on Fig. 8.

As for the $S/V$ ratio (Figs. 8c-d), we note for Dome C a slight discontinuity of the medium chord length decrease with density at approximately $800\,\mathrm{kg.m^{-3}}$. At this density, the medium chord length ceases to decrease. This discontinuity is difficult to interpret but may be linked to the pinching of the thinnest channels that arises at the start of the pore closure.

Figure 9b illustrates the ratio of the calculated chamber over throat pore size distribution for the 90th centile. Chamber pore size distribution is worked out using the granulometry over the whole sample, and the throat pore size distribution is obtained here by porosimetry on all sample faces, in the $X$, $Y$ and $Z$-directions. Figures 9B and 9II clarify the throat and chamber concepts (Sweeney and Martin, 2003). The porosimetry consists of the introduction of a sphere from the top surface, thus mimicking the intrusion of liquid from the specimen surface. Each tested size of spheres in the sample is associated to the volume fraction covered by the set of spheres. In Fig. 9II, this is represented by a light blue area in which a particular sphere can circulate. When the space between pores is too small the sphere cannot move down anymore. This is different from the chamber pore size distribution, which is only based on the geometry of pores. Spheres of different sizes are placed in every available space of the porous network in that case. We chose to display the distribution at the 90th centile, as the porosimetry is difficult to perform for deep sample. This provides the general trend of the closure of pores by pinching while still having large globular parts. Error bars show the standard deviations from the mean value of the porosimetry results on the six cubic faces. These are shown only for Dome C. After the air isolation density, there is a drastic increase of this ratio, as pinching reduces the throat size.

Percolation paths on sub-volumes were calculated in the $Z$-direction alone. They are evaluated by moving the largest possible sphere inside the pore network, from the top surface to the bottom one. For each sample, 10 percolation paths were calculated,





**Figure 9.** Variations with density of (a) the medium chord length, (b) the ratio of chamber size over throat size distribution at the 90th centile, (c) the maximal path diameter and (d) the connectivity index. The black dashed line is the average air isolation density ($\rho = 840$ kg.m$^{-3}$) from Martinerie et al. (1992) for Dome C. 3D images illustrate each parameter on the right hand side (images A-D). Images A-B-C come from the same DC-C12 volume at 85 m depth, while image D is a picture of a 91 m deep volume. $\phi_s$ is the diameter of the sphere which is used for visualization of porosimetry and granulometry. $\langle\phi_{max}\rangle$ is the mean of the maximum diameter for the different percolation paths. The last column (I-IV) sketches the calculation of parameters.





starting with a minimum path diameter of $24\,\mu m$, and then increasing it by $24\,\mu m$ steps. Figure 9c shows the calculated maximal path diameters. The vertical lines represent the variability based on all paths that are different, and the dots are the mean of those maximum diameters. A general decrease is observed, from approximately $300\,\mu m$ at $550\,kg.m^{-3}$ to zero (i.e. less than $24\,\mu m$ when reaching approximately $830\,kg.m^{-3}$, which is slightly lower than the air isolation density ($840\,kg.m^{-3}$,

(Martinerie et al., 1992)). Again, there are no noticeable differences between Dome C and Lock In. Both sites exhibit a large variability of the maximal path diameter that can vary between 0 and $200\,\mu m$ in the same sample. However these diameters are quantitative information for the path of air (in a given interval). A noticeable issue of this calculation is that the maximal path diameter depends on the size of the chosen sub-volume. Here a cube of $6.72\,mm$ size was used, which ensures percolation until $830\,kg.m^{-3}$ according to Fig. 9. The length of the path ranges from 8 to $18\,mm$, meaning a tortuous structure whatever

the depth. This is shown on image A in Fig. 9. A larger cube would have led to a drop of the maximum path diameter to 0 before $830\,kg.m^{-3}$ because of the structural isotropy of the pore network and the erratic process of closure. A smaller porosity amount is associated with a smaller probability of having a network connecting two opposite sample sides. Figures 9b-c do not allow for a clear differentiation between the DC-C12 and LI-C12 samples.

It is instructive to compare the evolution of the connectivity index in Fig. 9d to those of the Chamber/Throat and maximal

path diameter (Fig. 9b-c). The air isolation density obtained by Martinerie et al. (1992) for Dome C is well correlated to the end of the drop of the connectivity index (black dashed line). It also separates correctly the two regimes observed in Figs. 9b-c.

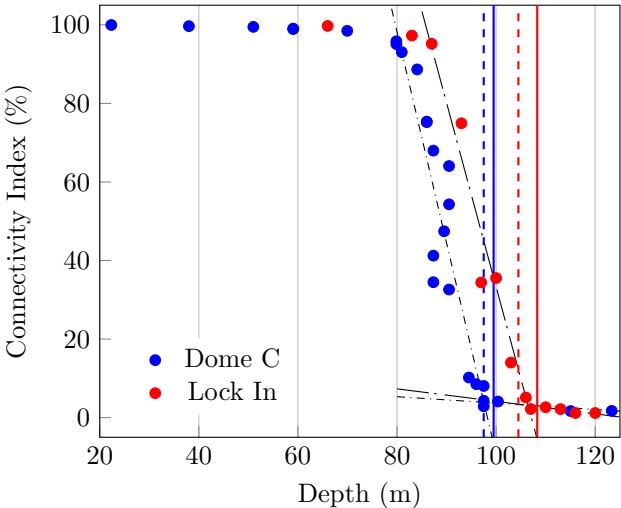

**Figure 10.** Connectivity index for Dome C and Lock In. Vertical solid lines represent the COD defined by the last firn air pumping depths in Table 1. Vertical dashed lines correspond to the air isolation depth for temperature of Dome C ($-55\,°C$) and Lock In ($-53.15\,°C$) calculated with the Goujon et al. (2003) parameterization of the air content related close-off density, and using the density profile with depth given in Fig. 3. Black dash-dotted lines are linear slopes of the connectivity index obtained by a least square method.

Figure 10 compares the evolution of the connectivity index with depth for Dome C and Lock In. When the connectivity index drops to very small values, Fig. 10 clearly shows a sharp change in the slope for the two sites. The two linear portions of the curve intersect at depths that are in good agreement both with the ultimate depth where air could be sampled and with



the Goujon et al. (2003) parameterization of air isolation depth related to air content data of Martinerie et al. (1992, 1994). The parameterization is used for temperatures of Dome C and Lock In.

Indeed, the blue and red solid lines correspond to the ultimate air pumping depths at Dome C (99.5 m) and Lock In (108.30 m) (see Table 1), whereas the blue and red dashed lines correspond to the parameterization of the air isolation density related
5   to air content measurements, called close-off density in Goujon et al. (2003).The air isolation depth was calculated from our measurements of density (Fig. 3). Using the connectivity index evolution thus seems a promising method to locate the depth/density at which pores are nearly fully closed. Its main advantage is that it does not depend on assumptions or arbitrary choices on the status of pores (closed or open). According to Fig. 10, the end of the connectivity index drop down at Lock In is approximately eight meters below the one of Dome C, consistently with the difference in last firn air sampling depths between
10   these sites.

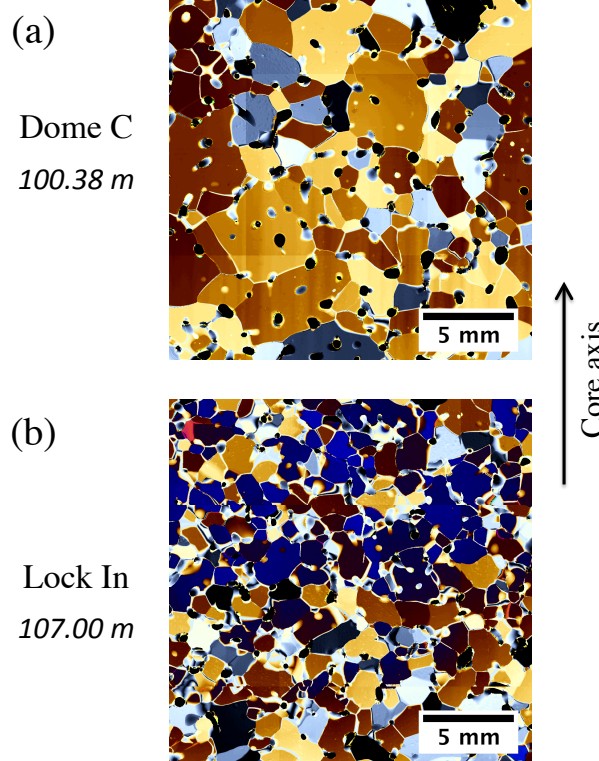

**Figure 11.** Comparison for the same density ($\rho = 851\,\mathrm{kg.m^{-3}}$) of (a) Dome C and (b) Lock In microstructures as observed by polarized light. Both thin sections are taken parallel to the core axis.

Although exhibiting a size distribution, grain size measurements showed a larger mean grain size at Dome C than at Lock In. Near the close-off depths of Dome C (100 m) and Lock In (108 m), the mean cross-sectional area was measured from thin sections to be approximately $1.78\,\mathrm{mm^2}$ and $0.59\,\mathrm{mm^2}$, respectively. We observed that grains in the Dome C firn increased in



size with depth (from $0.57\,\mathrm{mm}^2$ at $60\,\mathrm{m}$ to $1.78\,\mathrm{mm}^2$ at $100\,\mathrm{m}$) while grains in the Lock In firn showed no clear increase in size in the probed depth range (60 - 120 m). Figure 11 compares for the same density two thin sections of firn from Dome C and Lock In. It illustrates the clear difference in grain size between these two sites. This information on grain size, together with those on medium chord length (Fig. 9) and surface-area-to-volume ratio (Fig. 8) suggest a more tortuous pore structure

for Lock In.

## 7   Concluding remarks

X-ray characterization of firn samples was performed for two polar sites, Dome C and Lock In, over a broad depth range. We focus on the microstructural markers that accompany the closure of pores. The most obvious method to characterize pore closure is to measure the ratio of closed to total porosity. However, building on a thorough estimation of the results accuracy

after imaging and processing, we have shown that this approach has serious drawbacks as the percentage of closed porosity exhibits large variability with sample size, voxel size, and spatial heterogeneities when characterized by X-ray tomography. This is problematic since most models of air transport in the firn such as Battle et al. (2011); Witrant et al. (2012); Buizert et al. (2012); Trudinger et al. (2013) rely on a diffusion coefficient that requires the knowledge of the evolution of the closed porosity with depth and/or density. Thus, these models depend on a correct parameterization of the volume fraction of closed

pores and there is no consensus on those that have been proposed (Schwander, 1989; Goujon et al., 2003; Severinghaus and Battle, 2006; Mitchell et al., 2015; Schaller et al., 2017). The cut-pore effect (Martinerie et al., 1990) has also to be taken into account on the boundaries of the sample.

Here, we have encountered similar difficulties in determining unambiguously the fraction of closed pores. For example, we have determined that near the critical density of $840\,\mathrm{kg.m}^{-3}$, it ranges approximately from $20\,\%$ to $80\,\%$. This uncertainty

results from the conjunction of three detrimental effects: the sample size, the voxel size (resolution), and an inherent spatial heterogeneity.

We propose an alternative parameter: the connectivity index, which is simply the volume of the largest pore divided by the total pore volume. The main advantage of the connectivity index is that it is much less sensitive to cut-pores that undermine the measure of the closed porosity ratio. Before the close-off depth (COD), the connectivity index is essentially affected by

horizontal spatial heterogeneities in the sample but not much by resolution and sample size. When focusing on the determination of the COD, we have shown that a sample size of the order of $1$-$2\,\mathrm{cm}^3$ is a reasonable choice for the connectivity index determination. In these conditions, the connectivity index accurately predicts the COD as defined by the ultimate air sampling depth for the Dome C and Lock In sites. It also agrees with the parameterization of Goujon et al. (2003) related to the air content measurements from Martinerie et al. (1992, 1994). When using tomographic data, we propose to use the connectivity

index to determine precisely the COD.

That being said, the two sites studied in this work (Dome C and Lock In) are both characterized by cold temperatures and low accumulation rates. It would be interesting to use the connectivity index on other polar sites to confirm its relevance for





COD determination. Indeed, further challenging this index on polar sites exhibiting layering such as WAIS Divide, would help in determining if this metric is appropriate to characterize closure.

In comparison to our work, Schaller et al. (2017) took advantage of X-ray scans on slices of 4 cm height and of the full core diameter to determine the status of subvolume pores (closed or opened). Section 4 discussed the effects of the voxel size by

comparing results from samples using 12 and 30 µm. The difference for the closed porosity ratio was less than 7 %. Therefore, results obtained by Schaller et al. (2017) with a voxel size of 25 µm should not be too much vitiated by voxel-related errors. However, their definition of a closed pore still necessitates the knowledge of the status of the surrounding pores. This requires having access to larger volumes than those studied. In our work, only three samples have the diameter of the core. Besides, as detailed in section 4, the voxel size of the largest samples has a great influence on the fraction of closed pores. The large

samples and voxel size used by Schaller et al. (2017) seem appropriate to reasonably limit cut-pore effect without suffering too much from voxel size issues. However, it is still hindered by the unavoidable possibility of having a large closed pore (larger than the sample) that is wrongly considered open. The use of the connectivity index avoids these issues.

The morphology of pores was compared to data from Gregory et al. (2014) for polar sites WAIS Divide and Megadunes. Their data show more variability than ours, which can be explained by the selection of fine/coarse grained layers and the

different nature of the studied sites. Indeed, Dome C and Lock In are very cold sites with low accumulation, with Dome C being at the tip of the dome. In contrast, Megadunes is a site exposed to strong winds that redeposit snow from one side of the dune to the other, thus leading to a non-steady state site. WAIS Divide is a warmer site with high accumulation rate. Layering is significant, and high-density layers of fine grains can be found above low-density layers of coarse grains. As Gregory et al. (2014) intentionally extracted samples both from coarse and fine-grained layers, and plotted indiscriminately their data with

bulk density, these led to more dispersed results. Using a local density (density of the scanned sample rather than the density of the whole sample) as done here, could have reduced the dispersion of their results as discussed by Mitchell et al. (2015).

The polar sites Dome C and Lock In were extensively compared. Shapes of pores, closed porosity and connectivity indices are evolving likewise with density. However, the closure of pores occurs deeper for Lock In than for Dome C (by 8 meters). Our results on the medium chord length, the surface-area-to-volume ratio and grain size suggest a more tortuous firn at Lock

In than at Dome C. This could have interesting implications on air diffusivity in the firn at these two sites. If Lock In is more tortuous, older air samples should be retrieved from firn air pumping.

## 8   Code availability

Numerical codes for image analysis were developed using the free software Fiji (Schindelin et al., 2012) and available plugins (e.g., Boulos et al., 2013; Doube et al., 2010) as well as Python 3.5 libraries. All of them can be provided upon request.



## 9 Data availability

All the tomographic data plotted in the figures will be uploaded on the open-access library World Data Center for Paleoclimatology from the National Oceanic and Atmospheric Administration. Raw 3D images analyzed in the present work are also available upon request.

*Author contributions.* Alexis Burr, Clément Ballot, Pierre Lhuissier and Armelle Philip carried out the X-ray scans. The cold cell was designed at IGE. Machining of samples and image processing and property calculations were performed by Clément Ballot and Alexis Burr. Patricia Martinerie coordinated the recent Lock In drilling program and contributed to the field work. Armelle Philip and Christophe L. Martin directed the project. All authors contributed in the interpretation of results and to the writing of the manuscript.

*Competing interests.* The authors declare no conflict of interest.

*Acknowledgements.* We warmly thank Xavier Faïn for his help in cutting slices of the Lock In ice core. Edward Ando from the 3SR laboratory, and Mathieu Bourcier are thanked for providing help during scans. Christophe Frau is thanked for machining the cold cell and Gregory Teste for providing an air dryer. We are grateful to Amaëlle Landais and Anais Orsi for providing the $\delta^{15}N$ related lock-in depths for Dome C and Lock In. We also deeply thank the field personnel at Dome Concordia that retrieved the ice cores for the VOLSOL project: Joel Savarino, Philippe Possenti and wintering of the Concordia station. Field personnels at the Lock In site are also deeply thanked
(Jérôme Chappellaz, David Colin, Philippe Dordhain, Philippe Possenti). Agence Nationale de la Recherche (ANR) via contract NT09-431976-VOLSOL is acknowledged for the financial support for acquiring the ice core sections at Dome Concordia. This work relies on the lock-in ice core drilling and scientific program funded by IPEV program No1153 and CNRS INSU/LEFE program NEVE-CLIMAT. Philippe Possenti is also thanked for performing the DC12 drilling. Financial support for DC12 comes from the French ANR programs RPD COCLICO (ANR-10-RPDOC-002-01). The Institute Polaire Paul-Emile Victor (IPEV) supported the research and polar logistics through the
program GLACIOLOGIE No. 902. Labex OSUG2020 and CEMAM are thanked for financial support for the micro-computed tomograph.



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
