# Peer review of "Pore morphology of polar firn around closure revealed by X-ray tomography"

_The Cryosphere, 2018_

## Referee Comment (RC1) · I. Baker (Referee) · 13 May 2018

Good, very thorough, interesting paper. I suggest publishing as is.
* * *

---

## Referee Comment (RC2) · J. Freitag (Referee) · 6 Jun 2018

General comments

The paper of Alexis Burr et al. deals with the important issue of pore close-off in polar firn. It presents direct measurements of the pore network on small firn samples retrieved at two cold and dry Antarctic sites using X-ray microcomputer tomography. The authors calculated pore structure related parameters from the volume images and performed a comprehensive investigation of different errors associated with CT-applications. They found different tortuous pore networks at different locations, and suggest an alternative parameter, the connectivity index, for the prediction of close-off depth and critical density in CT-applications instead of using closed to total porosity

ratios.

The paper is well written, precise and thoroughly in great detail and comes to important conclusions. In my opinion, it is suitable to be published in the journal The Cryosphere. Nevertheless, I would suggest some minor revisions. A shortening of the manuscript would increase the clarity of the main conclusions/results. I am wondering if on can skip some of the figures (4 or 8ef or 9bcd?)

Specific comments

(section 3 and 4): I am wondering if you can discriminate between the resolution and sample size effect on closed porosity calculations? Did you perform measurements like M60, M30, M12, L12 or L30, L60 in your notation, meaning: different resolution, same volume?

(section 4.1): I missed the discussion related to the layered character of polar firn: density variations in vertical direction are much higher than in horizontal direction. The layering is a result of different deposition conditions/densification. What is the effect of sample selection (on the very small scale) on density? Could this explain some of your differences? What do you mean with representative density? Representative for a certain layer? Or only for the sample itself? The layered character of firn density imposes huge fluctuation in almost all properties when the data are plotted against depth. Therefore, I would not expect a smooth curve in Paramter(depth)-plots

(Section4.3): It would be very instructive for the reader to learn about the connectivity index (CI). What is the index meaning? It is in some sense related to the open porosity (ignoring all cut and isolated pores) and therefore a kind of compliment to the closed porosity? And, in my view, I would think that CI is size dependent, at least for already separated pore clusters. That is quite easy to understand if you consider separated bubbles: here the bubble size (or the largest bubble) would be counted for CI. Fine grained – small bubble ice would have a smaller CI than coarse grained structures with in average larger bubbles...

(related to CI) What is the reason for CI giving no information about LID but closeoff-depth? Any ideas?

(Section 5) " Fig. 7b shows that all points fall approximately on a master curve. This supports the idea that the close-off arises on first approximation at a particular density." This is a quite interesting observation and you should highlight this (repeat it in the conclusions!) although it is probably hard to bring it into the context of the introduction of the connectivity index. In this figure you compare really firn from totally different temperature regimes. Mega dunes, WAIS and your very cold sites on the East Antarctic Plateau! What is your opinion about this result? Is the coherence of all the closed porosity ratios on one master curve only the result of inaccurate estimations? It would be great to calculate the CI for all the sites! Could you do this?

FIGURES:

Figure 9a: missing "z" before "direction" in legend 9a Figure 9b,B: missing "t" in "throats" in axis label

UNITS:

Unit of density should be written as kgm-3 instead of kg.m-3

---

## Author Response (AR1)

First we would like to thank both referees for the reading of our manuscript and their helpful comments to improve it. The modifications brought to the revised manuscript are highlighted in blue and are between quotes. Latexdiff was used to compute the difference between the original and the revised versions of the manuscript, therefore modifications also appear in blue in the manuscript dedicated for revisions only (see at the bottom of the pdf file). Note that the last comment of Johannes Freitag on units and missing letters has been taken care of, but these modifications are not marked in the aforementioned manuscript.

**Response to Dr Johannes Freitag :**

**RC**: *I* am wondering if one can skip some of the figures (4 or 8ef or 9bcd?)**

10

15

5

**AC**: We believe figure 4 is instructive as it shows shapes of pores, differentiating open from the closed ones according to the definition of the manuscript. It also shows different stages of pore separations, which is uncommon in published research. Figures 8.e-f propose an index, commonly referred to, in the case of snow. It allows comparing with other polar sites characterized in the literature, which is not possible with many metrics. Figure 9.b and c are interesting as they support the idea of a closure at a particular density with two additional metrics informing on the percolation stage. They also give more information and details for further use on pore morphology. Removing these graphical supports would also render the manuscript reading quite difficult. Thus, we prefer these figures to be kept.

20 **RC**: *I* am wondering if you can discriminate between the resolution and sample size effect on closed porosity calculations? Did you perform measurements like M60, M30, M12, L12 or L30, L60 in your notation, meaning: different resolution, same volume?

AC: As shown by Table 3, measurements were performed for M30 and M60 (namely same sample size, different resolutions). It proves that loss in resolution (larger voxel size) leads to increased closed porosity ratio at constant volume (the same observation can be drawn between M60 and L60). This means that pores are artificially closed when thresholding larger voxelized images. But such effects seem to be dependent of the depth, as there may be more critical pore channels in the range 87-98m than at 101m when many channels are already separated. In short, it is difficult to evaluate which of the resolution or the sample size has the greater impact on closed porosity calculations at each depth.

30 Note that such discrimination is also performed on smaller samples. S30 and M30 have the same resolution and different sample sizes, and show larger closed porosity ratio for the M samples. S12 and S30 have the same size and different resolutions but exhibit no discernable differences.

Furthermore, the configuration of our tomograph (size of the detector and position to source), made it impossible for us to scan large or medium samples with a voxel size of  $12 \ \mu m$ : there is no M12 nor L12. However, L30 could be estimated by

35 interpolating X-ray scans performed at  $30\mu m$  of voxel size. Such a method produces information, as part of the image is not scanned at this voxel size. It also means huge amounts of data. Therefore such reconstruction was attempted but not completely analyzed.

To avoid such questions from readers we propose to add page 6 line 8 :

40

"Due to the size of the detector, and the position of the cold cell with respect to the X-ray source, there are no M12, L12 or L30 that could have been scanned."

45 **RC**: I missed the discussion related to the layered character of polar firn: density variations in vertical direction are much higher than in horizontal direction. What is the effect of sample selection (on the very small scale) on density? Could this explain some of your differences?

AC: The discontinuous measurements performed in our work do not allow us to discuss the layered character of polar firn. It is because of constraints inside the tomograph that we used such samples (larger in the cylinder axis direction than in diameter).

Beware that the box plots presented in figures 3.b, 5.a or 6.a contain samples S taken at the exact same depth with the exact same height separated by only  $\approx 0.1$  mm. As one can observe on Fig. 3b, they exhibit relatively low variability (less than 2%) between one another. We performed the same sampling about  $\approx 0.1$  mm above those samples (in the M30 sample), and did NOT observe a particular increase or decrease of density.

5 However, such sampling was limited to the three M30. It was never our intention to discuss layering with such samples. Continuous measurements at the 0.1mm scale seems more appropriate for such an objective.

Nevertheless, Dome C is known to have a limited layering and our discrete number of samples clearly do not enable to see any.

We propose to add page 10 line 15 :

10

15

"The layered nature of polar firn has already been studied and discussed in the literature [Hörhold 2011, Freitag 2013a], and could be detected using X-ray scanning [Freitag 2013b, Gregory 2014]. Very cold polar sites such as Dome C with low accumulation usually exhibit limited layering [Landais 2006], however the recent work of Fourteau et al. [2017] does show layering at the centimeter scale for the low accumulation sites of Vostok and Dome C. Our sample shape and discontinuous measurements do not allow us to discuss such anomalies in polar firn."

**RC**: What do you mean with representative density? Representative for a certain layer? Or only for the sample itself?

- 20 AC: Assuming a 2.5% variability on density as exhibited on Fig.3b, density of samples S, M or L are within margin of errors. Therefore, samples S are representative of the density for the whole slice of the firm (the whole slice being limited by the diameter of samples L and their height) with respect to a relative error of  $\pm 10 \ kgm^{-3}$  for the particular case of samples from 94.5 m depth for instance. The very low variability between DC-S12 densities is most striking for the three DC-S12 from 98 meters depth (840, 842, 843  $kgm^{-3}$ ). For more information on the representative volume element for the density of firm, we recommend to read in further details chapter 3 paragraph 5 of [Burr 2017].
- 25 we recommend to read in further details enapter 5 paragraph 5 of [D

We changed the last sentence of section 4.1 by :

"In conclusion, according to the relatively low variability between samples in terms of density, DC-S12 are large enough to 30 estimate the density of a slice of firm of the same height within 2.5% errors."

**RC**: It would be very instructive for the reader to learn about the connectivity index (CI). What is the index meaning? It is in some sense related to the open porosity (ignoring all cut and isolated pores) and therefore a kind of compliment to the closed porosity?

AC: Closed porosity ratio and connectivity index are defined in this way:

$$CP = \frac{V_{CP}}{V_{pores}}$$

40

and

$$CI = \frac{V_{largest\_pore}}{V_{pores}}$$

45

During densification,  $V_{CP}$  increases more or less and then decreases during bubble shrinking stage, while  $V_{pores}$  (volume of all pores) continues to decrease. Therefore, we expect CP to evolve at a rate completely dependent of the closed pore definition (one of its weakness). On the contrary, as densification proceeds, the largest pore diminishes. It is the rate of its decrease compared to the reduction of  $V_{pores}$  which is interesting, not its value. It will tend to 0, whatever the firm. The pertinent information

5 is how it evolves. The trend of the CI in our case can be separated in three stages, with two horizontal plateaus and a steep drop. It is the last slope change that enables to find the close-off depth. It would be most intriguing that the CI did not evolve this way. Let us imagine that it goes up after the drop, and goes down again, we could have concluded on the presence of a layer of open pores.

CI is really a physical parameter, easy to understand but completely dependent of the measure, while the CI is more geometrical, but will illustrate accurately the pore separation stage. Note also that it is related to the open porosity only until the 10 beginning of the close-off. When reaching the range of depth critical for pore separation (between 87 and 98m in Dome C), it can be either open or close, and because of the size of the sample, it could be considered an open pore in S12, but a closed one in M30. We think it is appropriate to compare both parameters, but it is important to keep in mind that it is not the open porosity, particularly when the CI decreases.

15

**RC**: And, in my view, I would think that CI is size dependent, at least for already separated pore clusters. That is quite easy to understand if you consider separated bubbles; here the bubble size (or the largest bubble) would be counted for CI. Fine grained - small bubble ice would have a smaller CI than coarse grained structures with in average larger bubbles.

20

AC: The CI is indeed strongly dependent of the statistical size of the pore. An unlikely very large closed pore inside a sample S would increase the CI while all pores are separated. Note that it is not sample size dependent as increasing the size of the samples does not affect the CI so much according to Fig.6 (compared to the closed porosity ratio).

It is important to understand that the evolution of the CI gives a trend on the closing process, but it is not self-sufficient. We believe that we also need morphological evolution of pores to complement it or to discriminate fine grained from coarse grained 25 structures. This is why we insist that parameters such as maximal path diameter, chamber and throat distribution, anisotropy or SMI are to be discussed and presented. Full information on pores is thus given.

Regarding the closed porosity definition we propose to simply add the following equation in the beginning of section 4.2:

**30**

"The closed to total porosity ratio (CP) is obtained by dividing the total volume of closed pores ( $V_{CP}$ ) in the ROI by the total volume of pores  $(V_{pores})$ :

$$CP = \frac{V_{CP}}{V_{pores}}$$

35

To improve the manuscript on the CI description we propose the following in the beginning of section 4.3 page 13:

"In this section we propose an alternative indicator of the pore closure, which is much less sensitive to the source of errors that characterize the closed porosity ratio, especially the sample size. The connectivity index (CI) is defined by the ratio between the volume of the largest pore  $(V_{largest pore})$  and the total volume of pores  $(V_{pores})$ : 40

$$CI = \frac{V_{largest\_pore}}{V_{pores}}$$

It was originally introduced by Babin et al. (2006) to depict void coalescence in bread. It proved useful (under the name of "interlinkage parameter") to quantify the coalescence of cavities during superplastic deformation of an aluminum alloy (Mar-45 tin et al., 2000), or as a criterion to optimize box sizes when smoothing density maps of graphite with inclusions (Babout et

al., 2006). The evolution of the CI leads to pertinent information on the percolation process of pores, as it describes how the largest pore evolves with the drop of the total pore volume. It tends to 0 when all pores are separated and/or shrinking. The trend is paramount for the CI interpretation, as fluctuations could mean unusual change in pore size distribution. Contrarily to the closed porosity ratio, the connectivity index is independent of sample boundary conditions, since all pores are considered. However, it is very dependent on the statistical size of pores and their number inside the samples."

And page 22 lines 18 :

"Figure 10 compares the evolution of the connectivity index with depth for Dome C and Lock In. When the connectivity
index drops to very small values, Fig. 10 clearly shows a sharp change in the slope for the two sites. As for Fig. 9.d, three stages can be clearly distinguished with two horizontal parts and an abrupt linear drop. The two last linear portions of the curve intersect at depths that are in good agreement with the ultimate depth where air could be sampled."

**15 **RC**: (related to CI) What is the reason for CI giving no information about LID but close off depth? Any ideas?**

AC: LID is usually defined by the stop of gravitational fractionation of δ15N [Battle 1996]. On a first guess we can assume that such fractionation needs a sufficiently low tortuosity and large channel diameter for percolation path, so that diffusion can occur. When LID is reached, the globular parts of pores would still be connected (at the resolution considered), giving a CI value located in the drop zone of the curve. This is true if we assume that air cannot diffuse on channel whose diameter are below 12μm (too tortuous).

RC: "Fig. 7b shows that all points fall approximately on a master curve. This supports the idea that the close-off arises on
first approximation at a particular density." This is a quite interesting observation and you should highlight this (repeat it in the conclusions!) although it is probably hard to bring it into the context of the introduction of the connectivity index. In this figure you compare really firn from totally different temperature regimes. Mega dunes, WAIS and your very cold sites on the East Antarctic Plateau! What is your opinion about this result? Is the coherence of all the closed porosity ratios on one master curve only the result of inaccurate estimations?

30

35

5

AC: Points fall approximately on a master curve because it is the density that rules the closure on first order. Note that it is the selection of samples from Gregory et al [2014] that leads to their scattered results. The choice of a local density may have move away their point from our Dome C and Lock In curves. Moreover, differences still exist in terms of the microstructure depending on the site, hence a full analysis of pore morphology. This may also have an impact on the scattering of points. This, combined with poor estimation of the closed porosity ratio leads to this approximate master curve. But as one knows, a little variation in the close-off density leads to very different  $\Delta$ age estimation...

We propose to discuss such behavior in the conclusion line 21 page 25:

40 "Using a local density (density of the scanned sample rather than the density of the whole sample) as done here, could have reduced the dispersion of their results as discussed by Mitchell et al. (2015). Such strategy could have helped confirm or reject that close-off occurs at a critical density as suggested by Schaller et al. 2017, while current inaccurate estimations do not allow such a claim. Again, we believe that the connectivity index would clearly answer this question, and would discriminate the role of the density and of the microstructure on closure if all curves fall on a master curve."

45

**RC**: It would be great to calculate the CI for all the sites! Could you do this?

AC: Such a parameter is very simple to compute, which is one of its strength, therefore we would only need the reconstructed tomographic volumes, or directly the volume of labelled pores. It would be interesting to compare the CI evolution for all the sites previously studied by X-ray tomography. This would allow the CI ability to estimate the COD to be confirmed.

5

**RC**: Figure 9a: missing "z" before "direction" in legend 9a Figure 9b,B: missing "t" in "throats" in axis label. Unit of density should be written as kgm-3 instead of kg.m-3.

AC: The revised manuscript takes care of these remarks.

**10**

20

[revised manuscript text omitted]
          | $2.5\mathrm{cm}$                   | -55                      | 3             | 99.5    |
| Lock In                   | 74°8.310′ S, 126°9.510′ E   | $pprox 4.5\mathrm{cm}$             | -53.15                   | $8 - 12^{c}$  | 108.3   |
| Dome Fuji bd   | 77°19′ S, 39°40′ E          | $2.1\mathrm{cm}$                   | -57                      | 0             | 104     |
| Vostok abe     | 77°28′ S, 106°48′ E         | $2.2\mathrm{cm}$                   | -57                      | 2             | 100     |
| WAIS Divide fg | 79°46.300′ S, 112°12.317′ W | 21 cm                              | -31                      | $\approx 10$  | 76.5    |
| Megadunes fh   | 80°77.914′ S, 124°48.796′ E | $< 4 \mathrm{cm}$                  | -49                      | $\approx 4$   | 68.5    |
| Summit ij      | 72°34.48′ N, 37°38.24′ W    | $21\mathrm{cm}$                    | -31                      | 10            | 80      |

*a* Bréant et al. (2017)

b Landais et al. (2006)

c Orsi (2017)

d Fujita et al. (2009)

e Barnola et al. (2004)

f Gregory et al. (2014)

g Battle et al. (2011)

h Severinghaus et al. (2010)

*i* Schwander et al. (1993)

j Witrant et al. (2012)

**3 Methods**

**3.1 Acquisition parameters**

X-ray micro-computed tomography was performed on samples coming from a large range of depths with a refined character-ization close to the depth at which closure of pores initiates. The majority of the sampling are samples labeled S (for small)
from Dome C (DC) and Lock In (LI), named DC-S12 and LI-S12 (see Table 2 for information on resolution of samples, which varies between 12 and 60 µm). S12 refers to small samples with voxel side length of 12 µm. These were drilled with a milling machine in slices of ice core such that the cylinder axis is along the core axis (vertical axis). The uncertainty of depth after drilling in slices is estimated to ± 0.05 m for all Lock In samples and most of Dome C samples (a few sample depths are known at ± 0.5 m). Samples were machined down to a diameter of 12 mm with a lathe. The machining operations were performed in

- a cold room at -10 °C, usually the day before scanning. Scanning geometry was helical in order to image long samples (from 20 mm to 30 mm) with a volume more than twice the value typically studied by Barnola et al. (2004) and Gregory et al. (2014). Scans were performed at 60 kV with 800 radiographs over 4 turns leading to a scan time of approximately 25 minutes per sample. Samples were positioned in a cold cell, cooled by air at -10 °C thanks to the coupling of an air dryer and a cryostat. The temperature was controlled by a thermocouple positioned against the sample-holder inside the cell. Schematic of the set
- 15 up and geometry of the samples are shown in Figs. 1 and 2.

Figure 1. A firn sample from Dome C (100.38 m) inside the cold cell during X-ray tomography imaging, with associated 3D images of a firn sample and of its pore network.

Three samples labeled L (for Large) were also characterized to investigate the scale effect. They were placed in polystyrene boxes with an eutectic cold pack inside. Samples and cold pack inertia kept the temperature below 0 °C, ranging from -18 °C to an observed maximum at -3 °C after 1.5 hours (corresponding to two scans). According to metamorphism experiments of Kaempfer and Schneebeli (2007) and considering the thermal inertia of these large samples, any microstructural evolution should be negligible during the 1.5 hour scanning time. Moreover, two samples were scanned a first time, kept at -10 °C for 6 months and then scanned a second time. No evolution was observed. Volumes M (for Medium) were also locally scanned inside the L samples, i.e. a second scan is performed at higher resolution (using a voxel size of 30 µm instead of 60 µm) and a larger acceleration voltage (80 kV instead of 60 kV). Due to the size of the detector, and the position of the cold cell with respect to the X-ray source, there are no M12, L12 or L30 that could have been scanned. Sample characteristics are detailed in Table 2 and the extracted volumes are illustrated in Fig. 2.

5

10

6